# Nanonization and Deformable Behavior of Fattigated Peptide Drug in Mucoadhesive Buccal Films

**DOI:** 10.3390/pharmaceutics16040468

**Published:** 2024-03-27

**Authors:** Woojun Kim, Hai V. Ngo, Hy D. Nguyen, Ji-Min Park, Kye Wan Lee, Chulhun Park, Jun-Bom Park, Beom-Jin Lee

**Affiliations:** 1College of Pharmacy, Ajou University, Suwon 16499, Republic of Korea; wj6619@ajou.ac.kr (W.K.); nvhai@ajou.ac.kr (H.V.N.); hynguyendinh@ajou.ac.kr (H.D.N.); djn05108@ajou.ac.kr (J.-M.P.); 2Dongkook Pharmaceutical Co., Ltd., Seoul 06072, Republic of Korea; lkw1@dkpharm.co.kr; 3College of Pharmacy, Jeju National University, Jeju 63243, Republic of Korea; chpark@jejunu.ac.kr; 4College of Pharmacy, Sahmyook University, Seoul 01795, Republic of Korea; 5Institute of Pharmaceutical Science and Technology, Ajou University, Suwon 16499, Republic of Korea

**Keywords:** mucoadhesive buccal film, formulation design, leuprolide-oleic acid conjugate, self-assembled nanonization, deformability index, enhanced permeability, in vitro evaluation

## Abstract

This study was tasked with the design of mucoadhesive buccal films (MBFs) containing a peptide drug, leuprolide (LEU), or its diverse nanoparticles (NPs), for enhanced membrane permeability via self-assembled nanonization and deformable behavior. An LEU-oleic acid conjugate (LOC) and its self-assembled NPs (LON) were developed. Additionally, a deformable variant of LON (d-LON) was originally developed by incorporating l-α-phosphatidylcholine into LON as an edge activator. The physicochemical properties of LON and d-LON, encompassing particle size, zeta potential, and deformability index (DI), were evaluated. MBFs containing LEU, LOC, and NPs (LON, d-LON) were prepared using the solvent casting method by varying the ratio of Eudragit RLPO and hydroxypropyl methylcellulose, with propylene glycol used as a plasticizer. The optimization of MBF formulations was based on their physicochemical properties, including in vitro residence time, dissolution, and permeability. The dissolution results demonstrated that the conjugation of oleic acid to LEU exhibited a more sustained LEU release pattern by cleaving the ester bond of the conjugate, as compared to the native LEU, with reduced variability. Moreover, the LOC and its self-assembled NPs (LON, d-LON), equivalent to 1 mg LEU doses in MBF, exhibited an amorphous state and demonstrated better permeability through the nanonization process than LEU alone, regardless of membrane types. The incorporation of lauroyl-L-carnitine into the films as a permeation enhancer synergistically augmented drug permeability. Most importantly, the d-LON-loaded buccal films showed the highest permeability, due to the deformability of NPs. Overall, MBF-containing peptide NPs and permeation enhancers have the potential to replace parenteral LEU administration by improving LEU druggability and patient compliance.

## 1. Introduction

Leuprolide (LEU) is a peptide drug functioning as a gonadotropin-releasing hormone (GnRH) receptor agonist used for the palliative treatment of prostate cancer, endometriosis, uterine leiomyomata, and central precocious puberty [1]. The physicochemical and biopharmaceutical properties of leuprolide (LEU) acetate are described in Table 1. Like many other therapeutic peptides and proteins, LEU faces significant challenges involving low oral bioavailability, owing to its susceptibility to first-pass metabolism. Furthermore, its hydrophilic nature and large molecular size contribute to its low permeability [2,3]. Because of these constraints, traditional oral administration is impractical, resulting in the need for alternative modes of delivery. Currently, LEU is primarily administered via intramuscular injection [4]: 3.75 mg once a month for patients with uterine leiomyoma and endometriosis and 7.5 mg once a month for patients with prostate cancer or central precocious puberty (weighing less than 25 kg) [5]. Unfortunately, patients who receive LEU injections often report discomfort and adverse side effects such as pain, burning sensations, hot flashes, headaches, and nausea. These side effects have been attributed to the reduction of hormone level after LEU administration [6]. Considering these limitations, there have been several parenteral formulations developed, including aqueous systems and controlled release systems for a long-acting LEU release [7,8,9,10]. However, these injectable formulations are still associated with pain and injection site reaction, limiting patient adherence [11].

In response to the challenges associated with the limitations of native LEU and conventional LEU injections, there is a growing imperative to explore and develop alternative routes of administration for LEU. Among them, non-invasive buccal drug delivery is considered a promising alternative route to enhance patient compliance, particularly in pediatric patients with central precocious puberty [12]. The buccal mucosa has good accessibility and physical robustness while bypassing hepatic and intestinal metabolism, making it a favorable route for delivering therapeutic peptides and proteins [13]. Moreover, the buccal mucosa is a well-vascularized tissue through which blood vessels drain directly into the jugular vein. Therefore, molecules capable of penetrating the buccal mucosal membrane are likely to be rapidly delivered into the systemic circulation [14], circumventing the hepatic first-pass effect and improving drug bioavailability [15]. Various buccal dosage forms, such as tablets, ointments, gels, films, and even nanocrystal-loaded formulations have been developed [16,17,18]. Among these, mucoadhesive buccal films (MBFs) hold distinct pharmaceutical advantages and demonstrate patient adherence due to their minimal side effects, greater flexibility than tablets, and precise dosing capabilities, as well as a larger surface area for drug absorption [19].

However, a significant challenge in the buccal delivery of LEU is to enhance its permeability in a controlled manner. To address this limitation, our research group pioneered the use of our fattigation platform to conjugate LEU with various fatty acids, increasing the lipophilicity of LEU and forming an amphiphilic chemical structure that could readily self-assemble into NPs [3,20]. Furthermore, we have also explored the use of deformable LEU NPs as a promising approach for elevating the permeability of LEU. Unlike conventional nanoparticles, deformable nanoparticles have unique properties that allow them to adapt to the dynamic environment of the buccal mucosa, which facilitates deeper penetration of the drug and better drug release [21,22]. It is known that the particle’s Young’s modulus, or the tensile stiffness of a material, is a critical physical property for vascular-targeted drug carriers [21]. As a result of their deformable behaviors, these NPs can pass through the tight spaces between the cells, ensuring efficient delivery of drugs through the membrane [23]. This innovative combination of self-assembled and deformable behaviors of LEU NPs holds great potential for optimizing and ultimately advancing the effectiveness of LEU delivery in a more patient-friendly and efficient manner.

Taken together, the aim of this work was to design MBFs containing LEU NPs with improved permeability and sustained LEU release, according to the mucoadhesiveness, self-assembled nanonization, and deformable behavior of fattigated LEU NPs, for noninvasive patient centricity. Hydroxypropyl methylcellulose (HPMC) and Eudragit^®^ RLPO were selected to prepare the MBFs. HPMC is a mucoadhesive polymer known for its ability to adhere to mucosal surfaces. In addition, it exhibits excellent film-forming properties and biocompatibility. On the other hand, Eudragit^®^ RLPO, a water-insoluble polymer, functions as an extended-release polymer with high permeability [24]. Initially, we investigated the optimal MBF formulations through a rigorous assessment of their physicochemical properties, in vitro dissolution, and permeability. Thereafter, LEU, LEU-oleic acid conjugate (LOC), LON, and d-LON, with or without adding lauroyl-L-carnitine as a permeation enhancer, were loaded onto the optimized MBFs to investigate the permeable performance of self-assembled NPs and deformable NPs, seeking to unlock their full potential for augmenting the membrane permeability and release rate of LEU. Intermolecular interactions between LEU and formulation ingredients in the films were confirmed using differential scanning calorimetry (DSC) and Fourier-transform infrared (FT-IR) spectroscopy. Finally, the surface morphology of the films containing LEU and NPs was confirmed using Field Emission Scanning Electron Microscopy (FE-SEM). 

## 2. Materials and Methods

### 2.1. Materials

Leuprolide acetate was purchased from Anygen (Gwangju, Republic of Korea). Eudragit RLPO was provided by Evonik Korea (Seoul, Republic of Korea). Hydroxypropyl methylcellulose (HPMC; Methocel K4M Premium EP) was provided by Colorcon Dow Chemical Company^TM^ (Dartford, UK). The propylene glycol was purchased from Samchundang Chemicals (Seoul, Republic of Korea). L-α-phosphatidylcholine and lauroyl-L-carnitine were purchased from Sigma-Aldrich (St. Louis, MO, USA). The permeation barrier membrane (PB-M) was purchased from Logan Instruments Corp. (Somerset, NJ, USA), and the PermeaPad membrane was supplied by InnoME GmbH (Espelkamp, Germany). Isopropyl myristate was used as a membrane activator (Sigma-Aldrich). HPLC-grade solvents, including anhydrous ethanol and acetonitrile, were purchased from Samchundang (Seoul, Republic of Korea). All other chemicals and solvents were of analytical or reagent grade and were used without further purification.

### 2.2. Preparation of LOC, LON, d-LON, and the MBFs

#### 2.2.1. Preparation of LOC, LON, and d-LON

The LOC and LON were prepared according to the experimental method described previously [20]. The acetate form of LEU was converted into LEU to avoid any redundant conjugation with acetate. Briefly, oleic acid (25 mM) was dissolved in 4 mL of THF and stirred at 700 rpm until completely dissolved. Triethylamine (14 μL) was then added, followed by the addition of benzoyl chloride (12 μL) and DMAP (1.22 mg). The reactive oleic acid-DMAP was replaced by the hydroxyl group of LEU to form LOC. The Yamaguchi esterification method with 4-dimethylaminopyridine (DMAP) and benzoyl chloride allowed the conjugation of oleic acid to the hydroxyl group of LEU. The synthesized LOC was purified using preparative HPLC (Prep-HPLC).

To convert LOC into self-assembled LON, 10 mg of LOC was dissolved in 2 mL of absolute ethanol. Then, 10 mL of deionized water (DW) was slowly added and stirred at 700 rpm at room temperature (25 °C) overnight. LON was obtained after solvent evaporation. In the preparation of d-LON, phosphatidylcholine was added to ethanol solution in the process of making LON. The ratio of the drug to phosphatidylcholine was 10:1 (*w*/*w*) [25].

#### 2.2.2. Preparation of MBFs 

MBFs were manufactured using the solvent casting method, with the formulation compositions of LEU-loaded MBF detailed in Table 2. To prepare the HPMC solution, a predetermined quantity of HPMC was added to a glass vial containing DW and allowed to swell for 5 min at 15,000 rpm using a homogenizer (HG-15D, Dai Han Scientific, Seoul, Republic of Korea). Simultaneously, LEU or its conjugate (LOC) or nano drug (LON, d-LON) was dissolved in a glass vial containing absolute ethanol under magnetic stirring at 700 rpm. While maintaining the stirring, Eudragit RLPO was gradually introduced into the aforementioned drug solution and dissolved. Subsequently, the drug/Eudragit RLPO solution was added to the HPMC solution, and propylene glycol was added and mixed for 90 min using a magnetic stirrer. The amounts of ethanol and DW were maintained as constant in all formulations. The final solution was then poured into the Teflon tray (98 × 70 mm) and dried in a Forced Convection Oven (OF-21E, Jeio Tech, Seoul, Republic of Korea) at 40 °C for 6 h. After drying, the films were cut into squares (1.5 cm^2^). 

### 2.3. Physicochemical Characterizations of LON and d-LON

#### 2.3.1. Dynamic Light Scattering (DLS)

The particle sizes and zeta potentials of LON and d-LON were measured using DLS (ELSZ-2000S, Otsuka Electronics, Osaka, Japan). Each sample was diluted five times with DW. All data were analyzed in triplicate, and the average values were determined.

#### 2.3.2. Deformability Index (DI)

The deformability of d-LON was assessed using a mini-extruder (Avanti Polar Lipids, Alabaster, AL, USA) and a syringe pump (NE-4000, New Era Pump Systems Inc., Farmingdale, NY, USA). A constant pressure of 15.55 mL/h was applied for 2 min to extrude d-LON through 100 nm polycarbonate membranes. The size of the d-LON was measured using DLS, as previously mentioned. The deformability index (DI) of the d-LON was calculated using Equation (1) [26]:(1)$$DI=J\times {\left(\frac{r_{v}}{r_{p}}\right)}^{2}$$
where *J* is the ratio of penetration through the permeable membrane, *r_v_* is the average size of the NPs after extrusion, and *r_p_* is the pore size of the permeable membrane (100 nm).

### 2.4. Physicochemical Characterizations of MBF

#### 2.4.1. Weight and Thickness Variations

The films (1.5 cm^2^) were weighed using a digital balance (GR-200, A&D Company, Tokyo, Japan), and the film thicknesses were measured using a thickness gauge (2109S-10, Mitutoyo, Kawasaki, Japan). All samples were measured in triplicate, and the averages and standard deviations were calculated.

#### 2.4.2. HPLC Analysis

Both LEU and LOC were simultaneously quantified using an HPLC system (Agilent 1200, Agilent Technologies, Santa Clara, CA, USA) equipped with a reverse-phase column (Poroshell 110 Å, C18 column, 250 × 4.6 mm, 5 Å, Phenomenex Gemini Å). The analysis was conducted using an isocratic method, with two distinct mobile phases, labeled A and B, dedicated to the determination of LEU and LOC, respectively. Mobile phase A was prepared by dissolving 10 mM sodium hexanesulfonate in a mixture of acetonitrile and deionized water (32:68, *v*/*v*), with the addition of 0.1% TFA. Conversely, mobile phase B was produced by incorporating 10 mM sodium hexanesulfonate into the mixture of acetonitrile and deionized water (68:32, *v*/*v*), with 0.1% TFA in order to produce mobile phase B. The column temperature was maintained at 25 °C, the flow rate was set to 1.0 mL/min, and the injection volume was controlled at 30 µL. For the detection of both LEU and LOC, the UV-VIS wavelength was set at 220 nm.

#### 2.4.3. Drug Content

The drug content in the buccal film was determined as follows: three films, each measuring 1.5 cm^2^, were dissolved in their respective mobile phases and sonicated for 10 min to ensure complete dissolution. After that, the solutions were filtered by a 0.45 μm PTFE syringe filter (Whatman, Maidstone, UK) for HPLC analysis.

#### 2.4.4. Folding Endurance

Folding endurance serves as an indicator of the film’s flexibility and resistance to wear. During the test, the film was subjected to manual folding of 180° at the same location until it eventually broke. The number of folding cycles before rupture occurred was counted as the folding endurance [27]. This test was conducted three times for each formulation (*n* = 3).

#### 2.4.5. Surface pH

To conduct the surface pH measurements, the films were initially wetted with 1 mL DW and left at room temperature for 2 h. Following this equilibration period, the surface of the films was brought into contact with the electrode of a pH meter (A211, Thermo Scientific, Waltham, MA, USA). Each sample was measured three times (*n* = 3).

#### 2.4.6. Swelling Study

The swelling properties of the polymeric film are essential for initiating mucoadhesion. The swelling properties of the films were evaluated using the swelling index (SI). The films were initially weighed and then placed on a Petri dish containing 20 mL of pH 6.8 buffer solution. At specific time intervals (15, 30, 60, 90, 120, 240, and 360 min), the swollen films were carefully dried using tissue paper to remove excess surface water. The films were then weighed again. The SI of the films was calculated using Equation (2) [28] (*n* = 3):(2)$$SI\;\left(\%\right)=\frac{{(W}_{t}-W_{1})}{W_{1}}\times 100$$
where *W*_1_ is the initial weight of the film, and *W_t_* is the weight of the film after each specified time interval.

#### 2.4.7. Field Emission Scanning Electron Microscopy (FE-SEM)

The surface morphology of the film was observed by FE-SEM (JSM-7900F, JEOL, Tokyo, Japan) at 3.0 kV. The films were coated with platinum for 2 min before measurement.

#### 2.4.8. Fourier-Transform Infrared (FT-IR) Spectrometer

The FT-IR spectra of the ingredients of the buccal films (Eudragit RLPO, HPMC, and propylene glycol), drugs (LEU, LOC, LON, and d-LON), and drug-loaded buccal films, as well as their physical mixtures, were analyzed using an FT-IR spectrometer (Nicolet iS550, Thermo Fisher, Tokyo, Japan) to discern any chemical interactions occurring between the drug and other pharmaceutical ingredients. The spectrum was scanned from 400 cm^−1^ to 4000 cm^−1^, with a resolution of 2 cm^−1^.

#### 2.4.9. Differential Scanning Calorimetry (DSC)

DSC (GmbH-DSC 200 F3 Maia, NETZSCH, Selb, Germany) was used to measure the thermal behavior of the polymers (Eudragit RLPO and HPMC), drugs (LEU, LOC, LON, and d-LON), and drug-loaded buccal films, as well as their physical mixtures. All samples were analyzed by weighing 5 mg and then heating it at the rate of 10 °C/min from 0 °C to 180 °C under a constant nitrogen purge condition, with a flow rate of 50 mL/min.

#### 2.4.10. In Vitro Residence Time

A modified USP Apparatus type II (paddle) method was used to measure the residence time, with slight adjustments made based on the previous method [3]. Before the experiment, the PB-M was immersed overnight in an isopropyl myristate solution to mimic the oral mucosal environment [20]. Then, the pretreated PB-M was affixed at a height of 7.5 cm from the bottom of the dissolution vessel, and the buccal film was placed on the PB-M. The vessel was then filled with 900 mL of pH 6.8 buffer solution. During the experiment, the machine was maintained at a constant speed of 50 rpm and a temperature of 37 ± 1 °C. The time at which the film completely eroded or detached from the PB-M was recorded and defined as the residence time. This measurement was conducted in triplicate for each formulation (*n* = 3).

#### 2.4.11. In Vitro Dissolution Study

The in vitro dissolution study of the MBFs was performed using the USP Apparatus type II (paddle) method, with the inclusion of a basket sinker (40 mesh, 25.4 mm × 34.9 mm), in 200 mL of pH 6.8 buffer solution. A UDT-804 (LOGAN Instruments Corp., Somerset, NJ, USA) was operated under a temperature of 37 ± 1 °C and a rotation speed of 50 rpm. At specific time intervals (15, 30, 60, 90, 120, 240, and 360 min), 3 mL of the sample was withdrawn and filtered using a 0.45 μm PTFE syringe filter (Whatman, Maidstone, England) for further HPLC analysis.

#### 2.4.12. In Vitro Permeability

The in vitro permeability was assessed by Franz diffusion cell (DHC-6TD, LOGAN Instruments Corp., Somerset, NJ, USA), with a diffusion area of 3.8 cm^2^. The membranes (either PB-M or PermeaPad) were securely clamped between the donor and the receptor chamber. Prior to the experiment, these membranes were soaked in an isopropyl myristate solution overnight to activate and mimic the oral mucosa environment. The MBF containing LEU, LOC, LON, or d-LON was placed on the surface of the activated membrane and lightly wetted with 1 mL of pH 6.8 buffer solution, mimicking the normal saliva pH, because saliva has a pH normal range of 6.2–7.6, with 6.7 being the average pH. To investigate the effect of the permeability enhancer, 10% (5 mg) of the calculated film weight (50 mg) in the F2 formulation, which is equivalent to a 1.0 mg LEU dose, was used.

The receptor chamber was filled with a pH 7.4 buffer solution, used to mimic the human physiological pH, which contained 20% ethanol as a sink condition and then was continuously agitated at 600 rpm using a magnetic stirrer. The samples (5 mL) were withdrawn at predefined time points (30, 60, 90, 120, 240, and 360 min) for HPLC analysis and replenished with 5 mL of fresh buffer solution. The entire diffusion system was maintained at 37 °C during the experiment, and a jacket was used to protect the cells.

## 3. Results and Discussion

### 3.1. Physicochemical Characterization of LON and d-LON

The physicochemical properties of LON and d-LON, including the particle sizes, polydispersity indices (PDI), zeta potential values, and deformability indices (DI) of the LONs and d-LONs, are summarized in Table 3. The particle sizes of LON and d-LON are 251.57 ± 5.52 nm (PDI 0.263 ± 0.006) and 269.43 ± 5.46 nm (PDI 0.168 ± 0.059), respectively. The slightly larger particle size of d-LON compared to LON was attributed to the use of phosphatidylcholine distributed on the surface of NPs as an edge activator for enhanced deformability. In general, the NPs are considered stable when the zeta potential values exceed +25 mV or fall below −25 mV [29]. For LON and d-LON, the zeta potential values are 71.69 ± 2.01 mV and 63.69 ± 4.56 mV, respectively. As a result, the addition of phosphatidylcholine to the structure of NPs did not impact their stability.

The DI values of LON and d-LON are 9.76 ± 1.24 and 55.99 ± 5.33, respectively. These results confirm that the notably higher DI value of d-LON is mainly caused by phosphatidylcholine, serving as an edge activator. Edge activators can aid in squeezing NPs when they permeate the mucous membrane; thereby augmenting the penetration of NPs into deeper layers [30]. In addition, a DI value greater than 50 is considered effective for enhancing mucosal permeation, and d-LON meets this criterion [31].

### 3.2. Formulation Optimization of MBF

#### 3.2.1. Physical Properties of MBF

For visual reference, Figure 1 shows photographic images of LEU-loaded MBF (F2). Table 4 shows the physicochemical properties of MBFs, such as film weight, thickness, drug content %, folding endurance, and surface pH, by varying polymer ratios. The targeted weight of the film was approximately 50 mg, depending on the weight of LEU and the structure of NPs. All formulations slightly exceeded the target weight, due to moisture content within the film. Formulation F5, composed solely of HPMC, exhibited the highest standard deviation of ±12.14, likely due to ingredient inhomogeneity attributable to the high viscosity of HPMC. Generally, the ideal thickness range of a buccal film is 50–1000 μm [32]. The thickness of the prepared films F1 to F5 varied from 213.33 ± 22.36 to 312.10 ± 69.82 μm, indicating that all formulations meet the appropriate criteria.

The targeted drug content per film was 0.5 mg (1.5 cm^2^). The drug content percentages for all formulations were between 96.18 ± 16.96 to 108.85 ± 8.60%, and the water content (LOD) invariably ranged 4.79 ± 0.39% after the drying process. Among formulations, F2 showed the lowest standard deviation, indicating superior reproducibility and stability in room temperature (25 °C) for one month, maintaining drug content, water content, and dissolution profiles. Conversely, F5 had the lowest drug content, which can be attributed to the non-homogeneous distribution caused by the high viscosity of HPMC, as mentioned earlier.

When the folding endurance of the film exceeds 300, it means that the film has a high level of mechanical strength and flexibility, allowing it to withstand breaks during administration and to remain in place at the application site for a long period of time [14]. All formulations endured more than 300 folds, confirming their robustness. Furthermore, the surface pH of the film is considered suitable between 4 and 8 in order to avoid irritation to the buccal mucosa [18]. All films remained within this desirable pH range, meeting this criterion.

#### 3.2.2. Swelling Study and In Vitro Residence Time

Rapid swelling of the film is important for achieving immediate and effective adhesion to the buccal mucosa [33]. Figure 2 shows the swelling indices for all MBF formulations associated with the different polymer ratios. F1, which contained only Eudragit RLPO, exhibited a 29.87 ± 12.28% swelling for 360 min. When comparing F2, F3, and F4, it was confirmed that the film with the highest HPMC content exhibited better swelling properties during the first 60–90 min than the films with the lower HPMC content. This rapid swelling is expected to facilitate effective attachment through quick interaction with the buccal mucosa. On the other hand, F5, comprising only HPMC, swelled to 116.20 ± 46.37% within 60 min but completely dissolved after 120 min, making further measurement impossible. It is believed that the rapid dissolution of F5 is related to the unique properties of HPMC, which, although capable of significant swelling at first, tend to disintegrate quickly due to its relatively low resistance to water penetration in an aqueous environment. Thus, the film structure was compromised, resulting in a dissolution of the film and a subsequent inability to evaluate its swelling properties.

To achieve optimal buccal drug delivery, it was reported that the adhesion time for the buccal film and the mucosa should be maintained within 4 to 10 h [34]. Table 5 shows the in vitro residence time of MBFs with different ratios of Eudragit RLPO and HPMC.

The film formulations such as F2, F3, and F4 exhibited appropriate attachment times. Notably, F2 had the longest attachment time (7.41 ± 0.55 h). In contrast, F5 displayed the shortest residence time, completely dissolving in the buffer in approximately 2 h. F1 detached after 3.74 ± 0.58 h, underscoring the need for HPMC in order to ensure proper film attachment, due to its higher viscosity grade [35].

#### 3.2.3. In Vitro Dissolution Study of Formulations

Figure 3 illustrates the dissolution profiles of LEU-loaded MBFs across 6 h and associated with varying polymer ratios (Figure 3a) and LEU doses ranging from 0.5 mg to 2 mg (Figure 3b), within the optimal F2 film formulation. Notably, formulations with higher HPMC contents demonstrated a lower dissolution rate within the first 90 min. Furthermore, films prepared by combining two polymers exhibited higher dissolution rates compared to films composed of only one polymer. Films F2, F3, and F4 exhibited dissolution rates exceeding 90% after 6 h, with F2 showing the highest dissolution rate at 97.31 ± 8.61%. Herein, the combination of Eudragit and HPMC appeared to have a synergistic effect, contributing to the increased dissolution rates [36]. In contrast, the changes of LEU doses in F2 film formulations did not significantly change the dissolution profiles. Although the film with a LEU dose of 1 mg had the highest dissolution rate, at 85.68 ± 17.25% after 1 h, the final dissolution rates of all films after 6 h did not exhibit significant differences. This suggests that varying the LEU dose within the F2 film formulation did not significantly impact the overall dissolution behavior of the film over the prolonged period of 6 h.

#### 3.2.4. In Vitro Permeability of Formulations

Figure 4 shows the in vitro permeability of LEU-loaded MBF, by varying polymer ratios (**a**) and LEU doses (**b**), for 6 h in buffer (pH 7.4), through PB-M using a Franz diffusion cell. Among the formulations in which two polymers were varied and combined, the permeability followed the order of F2 (13.87 ± 1.36%), F3 (13.27 ± 0.45%), F4 (10.15 ± 0.51%), and F5 (9.02 ± 1.42%) over 6 h (Figure 4a). This pattern suggests that a rapid increase in drug concentration, attributed to a high dissolution rate, enhances drug permeability.

The permeability for 6 h was also compared after changing the drug doses of the F2 formulation to 1 mg and 2 mg (Figure 4b). When the LEU dose was 0.5 mg, the drug was permeated at a rate of 13.87% for 6 h, while doses of 1 mg and 2 mg exhibited permeabilities of 21.73 ± 1.28% and 22.10 ± 0.69%, respectively. These results confirm that the drug dose affected permeability. However, there was no significant difference in permeability between the 1 mg or 2 mg drug doses. These findings suggest that up to a certain concentration level, higher drug concentrations enhanced drug permeability through PB-M [20].

### 3.3. Physicochemical Characterization of the Optimized MBF

Based on the above in vitro dissolution and permeability results, further studies involved the preparation of films loaded with LOC, LON, and d-LON using the F2 film formulation, in which the drug dose was increased from 0.5 mg to 1 mg. The drug amounts were adjusted to 1.22 mg for LOC and LON to match the 1 mg dose of LEU, while d-LON was used at 1.34 mg.

#### 3.3.1. Field Emission Scanning Electron Microscopy (FE-SEM)

Figure 5 shows the surface morphologies of different MBFs. The film exhibited a very smooth surface without the addition of drugs. However, the surface of the films became rough when the LEU, LOC, or NPs were added. However, all films exhibited homogeneous surfaces, without cracking or tearing, indicating that the addition of LEU, LOC, or NPs (LON or d-LON) did not significantly interfere with film structure. This observation underscores the robustness of the film formation process.

#### 3.3.2. Fourier-Transform Infrared (FT-IR) Spectrometer

Figure 6 shows the FTIR spectra of the pure LEU and pure pharmaceutical ingredients, as well as their physical mixtures and MBFs loading LEU, LOC, LON, and d-LON. The Eudragit RLPO spectra had specific peaks at 2993 cm^−1^ (CH) and 1727 cm^−1^ (C=O), and HPMC had a specific peak at 1057 cm^−1^ (C-O) [37]. The spectra of LEU showed peaks at 3269 cm^−1^ and 1630 cm^−1^, corresponding to the OH bond and C=O stretch peaks, respectively [33]. In addition, propylene glycol exhibited an OH peak at 3433 cm^−1^ [1]. The shift of the OH bond peak from 3269 cm^−1^ (LEU) to 3282 cm^−1^ (LOC, LON, and d-LON) and the C=O bond peak from 1630 cm^−1^ (LEU) to 1651 cm^−1^ (LOC, LON, and d-LON) were evidence of oleic acid conjugation [20]. In the d-LON spectra, the presence of phosphatidylcholine was indicated by peaks at 2921 cm^−1^ and 2851 cm^−1^ [38]. When comparing the spectra of each buccal film with the spectra of their physical mixtures, no new peaks were generated, and no significant peak shifts were observed. This confirms that no intermolecular interactions occurred between the pharmaceutical ingredients of the film, LEU, LOC, and NPs. Based on these results, it can be concluded that the films exhibit structural integrity and no chemical alterations occurred during the formulation process. 

#### 3.3.3. Differential Scanning Calorimetry (DSC)

Figure 7 shows the DSC thermograms of the film ingredients, their physical mixtures, and MBF loading LEU, LOC, LON, and d-LON. DSC data could provide the homogeneity and crystallinity of the drug and other ingredients in the film. LEU exhibited intrinsic endothermic peaks at 87.7 °C and 154.1 °C, representing the glass transition temperature (Tg) and the melting point of intact LEU [39]. The thermograms of LOC, LON, and d-LON showed similar patterns, while d-LON exhibited an additional peak at 131.5 °C. Endothermic drug peaks were also observed in all the physical mixtures; however, these peaks were not visible in the manufactured films containing LEU, LOC, and NPs, suggesting that these films are in an amorphous state.

#### 3.3.4. In Vitro Dissolution Study

Figure 8 shows the drug dissolution profile of MBFs containing LEU, LOC, LON, and d-LON for 6 h in a pH 6.8 buffer. The LEU-loaded buccal film exhibited the highest dissolution rate, reaching 98.61 ± 6.19%. The other films showed high dissolution rates in the following order: LOC (86.87 ± 7.33%), d-LON (82.00 ± 6.27%), and LON (80.07 ± 7.58%), for 6 h. Notably, the d-LON-loaded buccal film showed a slightly higher drug release than the LON-loaded buccal film after 6 h. This could be attributed to the presence of phosphatidylcholine, which acts as a solubility enhancer owing to its amphiphilic nature, leading to increased drug solubilization and, subsequently, a higher dissolution rate [40].

It is noteworthy that more than 80% of the drug was released from all films after 6 h. In addition, the film exhibited a sustained release pattern and lower variations in standard deviations when fatty acids were conjugated to LEU. It should be noted that the relatively higher standard deviations in the dissolution profile of LEU-loaded film may be attributed to the lower stability of native LEU in buffer solution. This controlled release behavior is promising for achieving improved drug delivery and prolonged therapeutic effects.

#### 3.3.5. In Vitro Permeability Test

Figure 9 shows the in vitro permeability of MBFs containing LEU, LOC, LON, and d-LON, for 6 h in buffer (pH 7.4) through PB-M or PermeaPad^®^ membrane using a Franz diffusion cell. PermeaPad^®^ is a biomimetic membrane used for simulating and investigating efficient passive permeability through different barriers in the body (e.g., GIT, buccal, and nasal) [41]. A PermeaPad^®^ barrier (an estimated cut-off size of 7–10 kDa) is hand-crafted by depositing phospholipids (phosphatidylcholine) and is widely used to predict buccal mucosal permeability due to the robustness of the membrane and stability with solvents and bio-relevant solutions. Permeation-Barrier Membrane (PB-M) is a cellulose membrane (pore size not applicable) having a similar acidity to the human epidermis, which allows the transfer of novel drugs. In the case of PB-M (Figure 9a), the permeability of the LEU-loaded buccal film showed the lowest value, at 21.73 ± 1.28% for 6 h. The LOC-loaded films exhibited improved permeability. The permeability of the LOC-loaded buccal film was 28.42 ± 0.79%, and that of the LON-loaded buccal film was 33.37 ± 1.84%. These values are 1.31 and 1.54 times higher, respectively, than that of the LEU-loaded buccal film. This increase in permeability can be attributed to the cis double bonds in oleic acid, which can induce kinking of the alkyl chains and hinder stratum corneum lipid packing [42,43]. Moreover, the smaller size of the NPs can enhance permeability through self-assembled nanonization [44]. Furthermore, the d-LON-loaded buccal film exhibited the highest permeability, which was 51.38 ± 1.64% for 6 h, thanks to its deformable behavior. This value is 2.36 times higher than that of the LEU-loaded buccal film. The d-LON and LON differ in surface elasticity owing to the addition of phosphatidylcholine as a deformable ingredient. This difference in elasticity allows d-LONs to permeate membranes with pore sizes smaller than their diameter without physical assistance [45]. When PB-M was replaced with PermeaPad^®^ membrane under the same conditions, the permeability decreased, but the overall permeability pattern remained similar (Figure 9b).

Our research team has reported that lauroyl-L-carnitine has a significant effect on the permeability improvement of mucoadhesive buccal tablets [3]. As a follow-up research effort, Figure 10 compares the in vitro permeability of MBFs containing LEU, LOC, LON, and d-LON with lauroyl-L-carnitine (5 mg, 10% in weight ratio of the film) for 6 h in a pH 7.4 buffer through PB-M or PermeaPad^®^ membrane using a Franz diffusion cell. In the case of PB-M, the permeability levels of the films containing the permeation enhancers were as follows: LEU-loaded buccal film (27.46 ± 0.27%), LOC-loaded buccal film (35.89 ± 1.20%), LON-loaded buccal film (45.44 ± 1.37%), and d-LON-loaded buccal film (62.35 ± 0.21%). When PermeaPad^®^ membrane was used, d-LON-loaded buccal film (33.22 ± 1.61%) exhibited the highest permeability, while LEU-loaded buccal film (8.23 ± 0.58%) had the lowest permeability with consistent permeability profiles. The permeabilities of all MBFs containing the permeation enhancer were absolutely improved, giving approximately 44% improvement in PB-M and 25% improvement in the PermeaPad^®^ membrane. Notably, the d-LON-loaded MBF containing the permeation enhancer achieved the highest permeability, as compared to the LEU-loaded MBF, with the aid of simultaneous self-assembled nanonization and deformable behavior. The reduction of drug particle size via self-assembled nanonization could also contribute to improved permeability [3,20]. It is worth mentioning that interactions between NPs and biological membranes are also among the most important phenomena for developing NP-based therapeutics and overcoming nanotoxicology in drug delivery, cancer targeting, and immune modulation [46].

While applying LON- or d-LON-loaded MBF in buccal mucosa, the nanoparticles diffused out and then experienced extensive hydrolysis by esterase in saliva or blood, liberating active LEU, as reported previously by our group [47]. It is well known that human saliva presents esterase activity, in addition to blood having abundant amounts of esterase [48].

## 4. Conclusions

MBFs loaded with LEU, LOC, LON, and d-LON were successfully prepared using a simple solvent casting method. The MBF formulation and LEU doses were optimally designed based on the mechanical properties, swelling indices, in vitro residence time measurements, in vitro dissolution tests, and in vitro permeation tests. The surface morphology of MBF appeared smooth, without showing any cracks or tears. The LEU, LOC, and NPs in the film were present in an amorphous state. In the in vitro dissolution test, all films exhibited a high dissolution rate of more than 80% for 6 h. Notably, the conjugation of oleic acid to LEU demonstrated a distinct advantage, with a sustained release pattern, in contrast to the native LEU. The in vitro permeability was the highest in the case of d-LON-loaded film containing permeation enhancer (62.35 ± 0.21% for 6 h), followed by, in order, LON, LOC, and LEU-loaded films. The current fattigation platform providing self-assembled nanonization and deformable behavior could be used to improve the LEU permeability via the buccal route. In addition, the MBF containing the permeation enhancer exhibited remarkable permeability improvement. Based on these findings, the conceptual combination of mucoadhesiveness, fattigation platform technology, and the deformable behavior of NPs could provide a research challenge which could enhance permeability and overcome the limitations of the buccal delivery of therapeutic peptides like LEU. In the future, in vivo experiments will be performed to validate the current in vitro results. Therefore, this study presents a promising and innovative approach for the buccal delivery of peptide drugs.

## Figures and Tables

**Figure 1 pharmaceutics-16-00468-f001:**
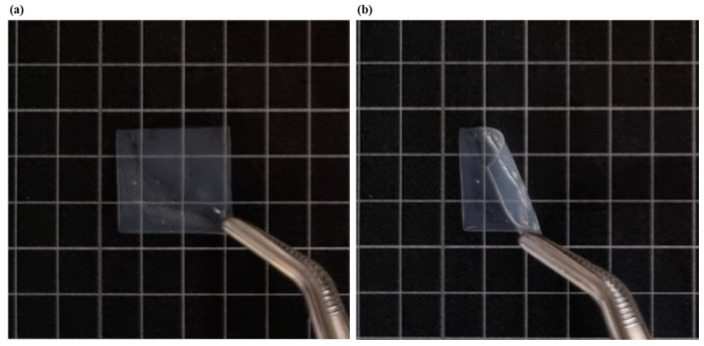
Photographic images of LEU-loaded MBF (F2): (**a**) before bending, (**b**) after bending.

**Figure 2 pharmaceutics-16-00468-f002:**
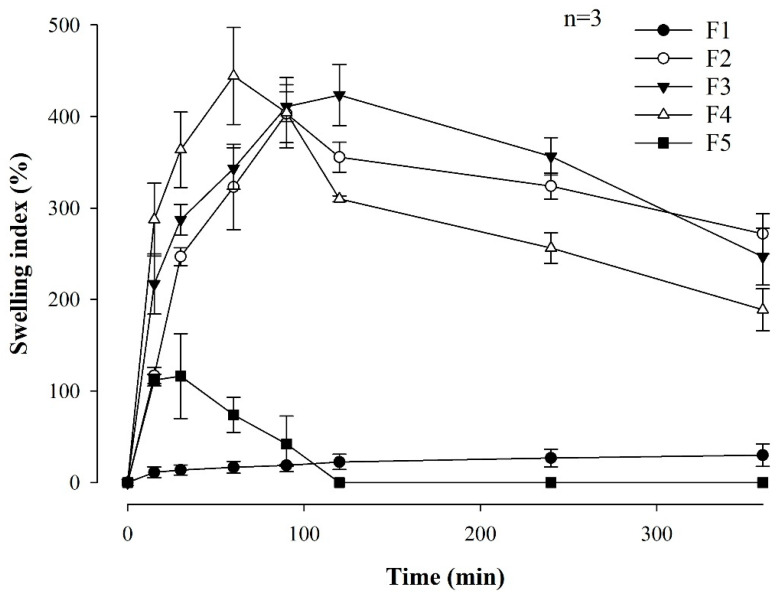
Swelling index of MBF formulations associated with the different polymer ratios (*n* = 3).

**Figure 3 pharmaceutics-16-00468-f003:**
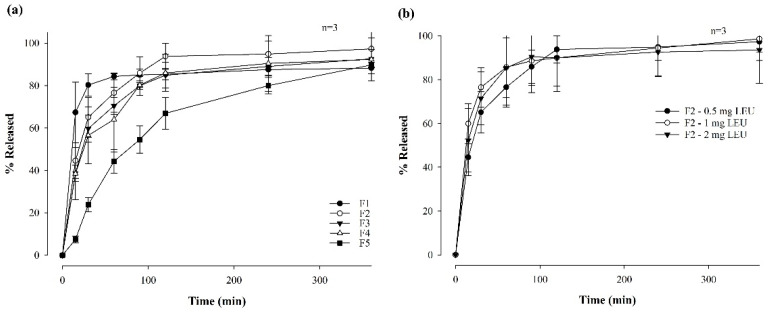
Dissolution profiles of MBF for 6 hrs in a pH 6.8 buffer by varying polymer ratios (**a**) and LEU doses (**b**) (*n* = 3).

**Figure 4 pharmaceutics-16-00468-f004:**
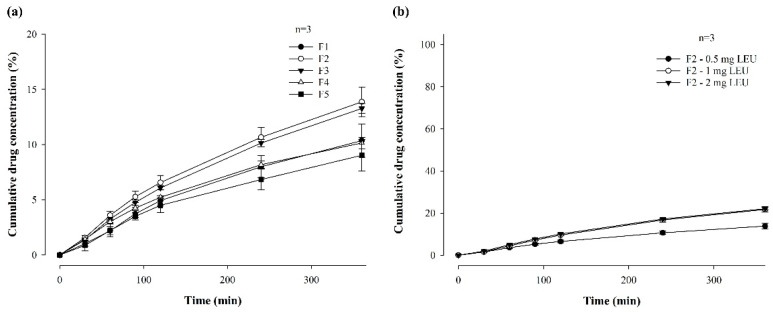
In vitro permeability of LEU-loaded MBF by varying polymer ratios (**a**) and LEU doses (**b**), for 6 h in buffer (pH 7.4), through PB-M using a Franz diffusion cell (*n* = 3).

**Figure 5 pharmaceutics-16-00468-f005:**
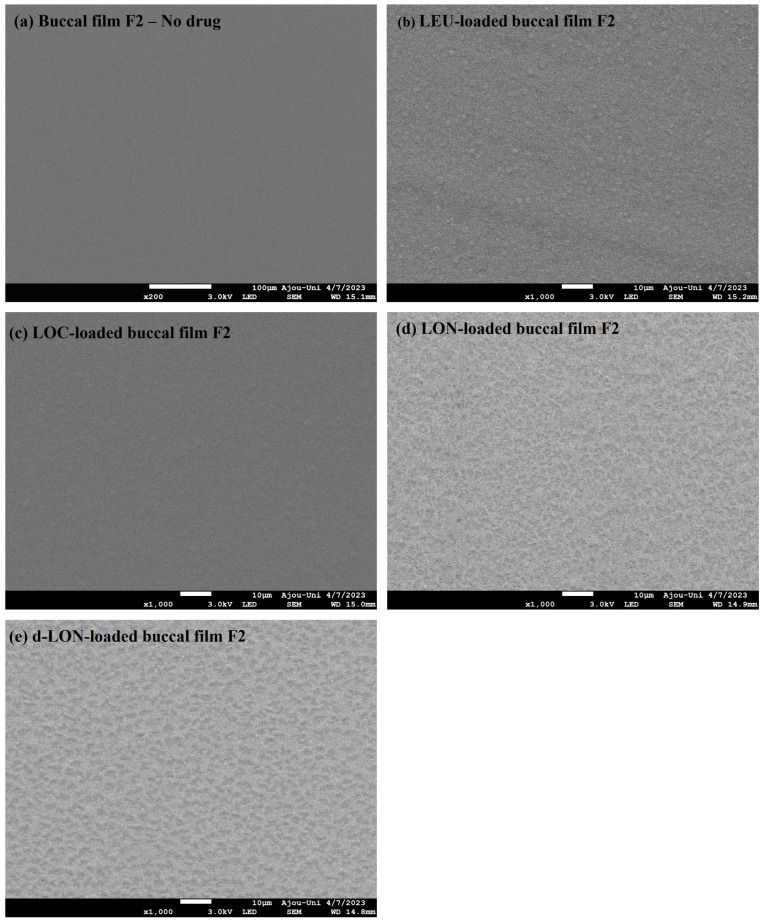
Surface FE-SEM images of various MBFs: (**a**) no drug (blank buccal film F2), (**b**) LEU-loaded buccal film F2, (**c**) LOC-loaded buccal film F2, (**d**) LON-loaded buccal film F2, and (**e**) d-LON-loaded buccal film F2.

**Figure 6 pharmaceutics-16-00468-f006:**
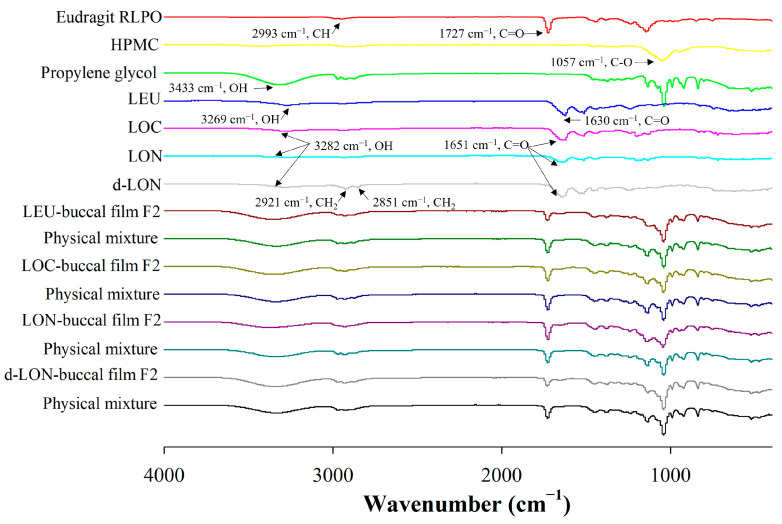
The FT-IR spectra of ingredients in the preparation of MBFs, LEU, LOC, LON, d-LON, LEU-loaded buccal film F2, LOC-loaded buccal film F2, LON-loaded buccal film F2, d-LON-loaded buccal film F2, and their physical mixtures.

**Figure 7 pharmaceutics-16-00468-f007:**
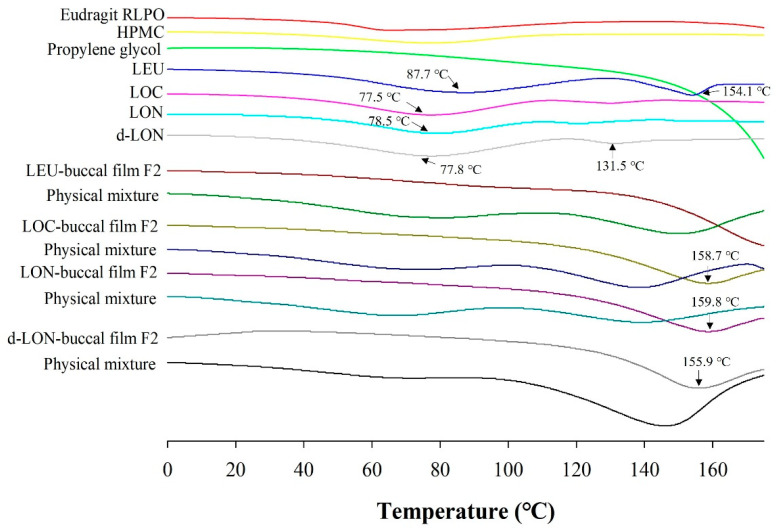
The DSC thermograms of ingredients in the preparation of MBFs, LEU, LOC, LON, d-LON, LEU-loaded buccal film F2, LOC-loaded buccal film F2, LON-loaded buccal film F2, d-LON-loaded buccal film F2, and their physical mixtures.

**Figure 8 pharmaceutics-16-00468-f008:**
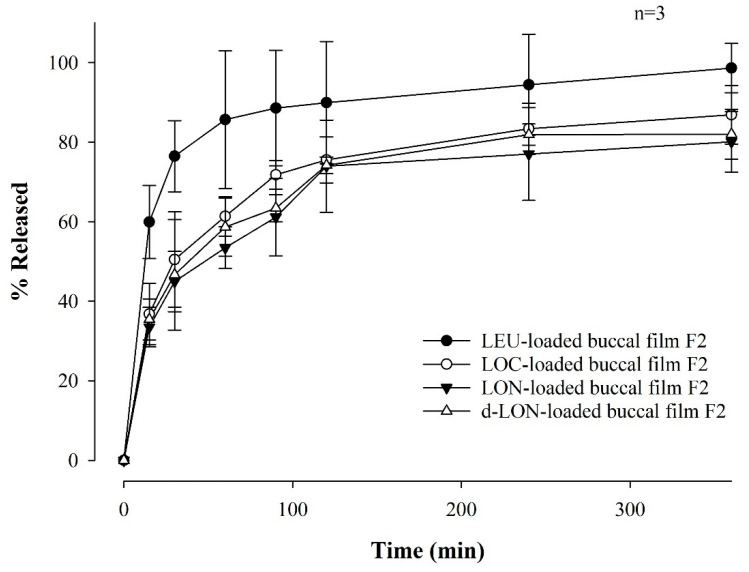
Dissolution profiles of MBFs containing LEU, LOC, LON, and d-LON, for 6 hrs in a pH 6.8 buffer (*n* = 3).

**Figure 9 pharmaceutics-16-00468-f009:**
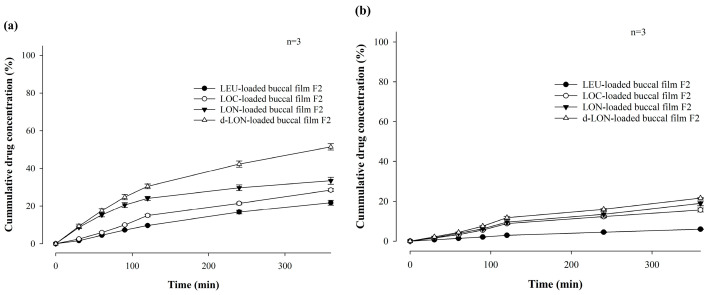
In vitro permeability of mucoadhesive buccal films containing LEU, LOC, LON, and d-LON (*n* = 3): (**a**) PB-M and (**b**) PermeaPad^®^ barrier.

**Figure 10 pharmaceutics-16-00468-f010:**
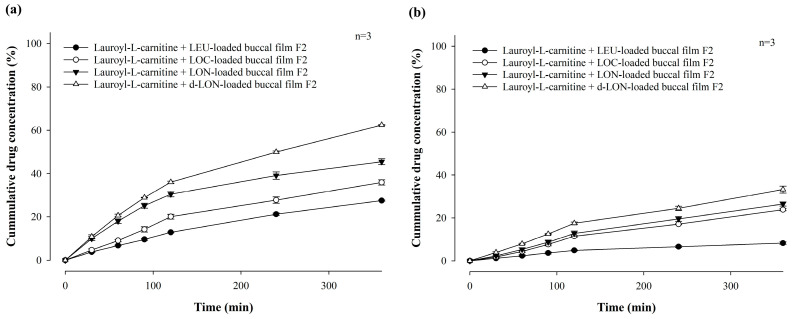
In vitro permeability of mucoadhesive buccal films containing LEU, LOC, LON, and d-LON with lauroyl-L-carnitine (*n* = 3): (**a**) PB-M, (**b**) PermeaPad barrier.

**Table 1 pharmaceutics-16-00468-t001:** Physicochemical and biopharmaceutical properties of leuprolide (LEU) acetate.

Factor	Leuprolide Acetate
2D structure	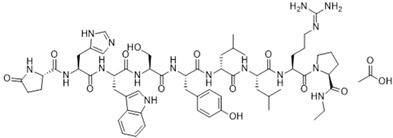
Physical appearance	White solid
Molecular weight	1269.45 g/mol
Chemical formula	C_59_H_84_N_16_O_12_ ∙ C_2_H_4_O_2_
Melting point	150–155 °C (https://www.chemicalbook.com/ChemicalProductProperty_EN_CB7174300.htm (accessed on 20 March 2024))
Half-life	3 h
Drug class	GnRH analogue; GnRH agonist; Antigonadotropin
Route of administration	IM, SC injection
Excretion	Kidney
Solubility in water	Highly water-soluble, ≥66.66 mg/mL
log P	−2.7
Dose strength	3.75, 7.5 mg per month

**Table 2 pharmaceutics-16-00468-t002:** Formulation compositions of LEU-loaded MBFs (unit: mg).

Code	Drug Type	Eudragit RLPO	HPMC	Propylene Glycol	Ethanol	DW
F1	0.5 (LEU)	21	-	28	0.3 mL	0.2 mL
F2 *	0.5 (LEU)	14	7	28	0.3 mL	0.2 mL
	1.0 (LEU)	14	7	28	0.3 mL	0.2 mL
	2.0 (LEU)	14	7	28	0.3 mL	0.2 mL
	1.22 (LOC)	14	7	28	0.3 mL	0.2 mL
	1.22 (LON)	14	7	28	0.3 mL	0.2 mL
	1.34 (d-LON)	14	7	28	0.3 mL	0.2 mL
F3	0.5	10.5	10.5	28	0.3 mL	0.2 mL
F4	0.5	7	14	28	0.3 mL	0.2 mL
F5	0.5	-	21	28	0.3 mL	0.2 mL

* F2 formulation was optimally chosen to prepare the film. The effect of the LEU doses in the film was investigated. LOC (1.22 mg), either LON (1.22 mg) or d-LON (1.34 mg) having 1.22 mg LON and 0.12 mg phosphatidylcholine, equivalent to 1.0 mg LEU dose, was loaded in the film.

**Table 3 pharmaceutics-16-00468-t003:** Physicochemical properties of LON and d-LON (*n* = 3).

NPs	Particle Size (nm)	PDI	Zeta Potential (mV)	DI
LON	251.57 ± 5.52	0.263 ± 0.006	71.69 ± 2.01	9.76 ± 1.24
d-LON	269.43 ± 5.46	0.168 ± 0.059	63.69 ± 4.56	55.99 ± 5.33

**Table 4 pharmaceutics-16-00468-t004:** Physicochemical properties of MBF with different polymer ratios (*n* = 3).

Code	Film Weight (mg)	Film Thickness (μm)	Drug Content (%)	Folding Endurance	Surface pH
F1	57.23 ± 6.48	246.87 ± 22.25	102.50 ± 18.83	>300	5.72 ± 0.18
F2	55.83 ± 2.57	222.97 ± 10.27	103.78 ± 4.21	>300	5.96 ± 0.08
F3	53.97 ± 5.15	213.33 ± 22.36	108.85 ± 8.60	>300	6.11 ± 0.11
F4	51.63 ± 6.84	271.93 ± 52.23	106.63 ± 14.21	>300	5.79 ± 0.38
F5	49.57 ± 12.14	312.10 ± 69.82	96.18 ± 16.96	>300	6.24 ± 0.16

**Table 5 pharmaceutics-16-00468-t005:** In vitro residence time of MBFs with different polymer ratios (*n* = 3).

Code	Residence Time (h)
F1	3.74 ± 0.58
F2	7.41 ± 0.55
F3	6.83 ± 0.30
F4	6.57 ± 0.47
F5	2.23 ± 0.29

## Data Availability

The original contributions presented in the study are included in the article, further inquiries can be directed to the corresponding author.

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
