# Peer review of "Nanonization and Deformable Behavior of Fattigated Peptide Drug in Mucoadhesive Buccal Films"

_pharmaceutics, 2024, doi:10.3390/pharmaceutics16040468_

Round 1
Reviewer 1 Report
Comments and Suggestions for Authors
This is a well-written paper, with detailed description of methods and clear, succinct presentation of results. The authors have published a series of papers on the topic, as cited in the article. Thay have already demonstrated the benefits of the LEU-oleic acid conjugate, its incorporation into self-assembled NPS and subsequent incorporation into mucoadhesive tablets; the permeation enhancer has already been reported.
The focus of this paper is therefore the novelty afforded by the use of deformable particles and the development of buccal films. My only concern is the lack of any stability considerations in the paper.
Abstract: With reduced variability rather than narrowed standard deviations. This is a comment throughout. It is the variability that caused the derived standard deviation to be larger or smaller.
Abstract states that d_LON is more permeable due to deformability. How has it been proven that this is the only reason?
Line 51. Are the side effects from the leuprolide or the formulation? The reference cited did not seem to contain this information. There have been several formulations developed including aqueous systems and controlled release systems. A short reference to this could be included.
A short explanation governing the selection of polymers for formulation of the films should be included.
Line 343, explain sufficient drug delivery to the buccal film. Does this mean release from the particles into the film, or is it expected that the nanoparticles diffuse intact. Is there any evidence of this?
There are no stability studies in the paper and the authors should comment on how any implications for storage of films and how stability could be affected. What was the water content of the different films? DSC does not seem to indicate that there is residual moisture, but how was this controlled? Given the documented stability issues with this drug (and indeed referenced in the dissolution testing), the authors could discuss any impact of modifications on the stability profile and the impact of this in formulations.
Author Response
Response letter to reviewers’ comments:
Reviewer #1: This is a well-written paper, with detailed description of methods and clear, succinct presentation of results. The authors have published a series of papers on the topic, as cited in the article. Thay have already demonstrated the benefits of the LEU-oleic acid conjugate, its incorporation into self-assembled NPS and subsequent incorporation into mucoadhesive tablets; the permeation enhancer has already been reported.
The focus of this paper is therefore the novelty afforded by the use of deformable particles and the development of buccal films. My only concern is the lack of any stability considerations in the paper.
|
Comments |
Responses |
|
1. Abstract. With reduced variability rather than narrowed standard deviations. This is a comment throughout. It is the variability that caused the derived standard deviation to be larger or smaller.
|
Thank you for your thoughtful suggestion. We agreed and revised this sentence in line 26.
|
|
2. Abstract states that d-LON is more permeable due to deformability. How has it been proven that this is the only reason? |
Many thanks for your excellent question. In Table 3, the size of d-LON was slightly larger than LON, however the deformability index (DI) value (15 times) of d-LON was significantly higher than LON and met the criteria for enhancing mucosal permeation. In addition, in vitro permeability test (section 3.3.5, line 501-511), we indicated that the d-LON-loaded buccal film exhibited the highest permeability, which was 51.38 ± 1.64% (without adding enhancers) for 6 h, thanks to its deformable behavior as compared to LON-loaded buccal film (33.37±1.84%) and LOC-loaded buccal film (28.42±0.79%). Furthermore, FTIR result confirmed that no interaction occurred between the pharmaceutical ingredients of the film and nanoparticles (section 3.3.2). Collectively, we confirmed that the deformable behaviors of d-LON were one of key factors for the enhanced permeation, as compared to LOC or LON.
|
|
3. Line 51. Are the side effects from leuprolide or the formulation? The reference cited did not seem to contain this information. There have been several formulations developed including aqueous systems and controlled release systems. A short reference to this could be included. |
We appreciate your valuable question. We are so sorry for making confusion. The side effects mentioned in this cited reference are from the low estrogen level after leuprolide administration. We revised and added the more information with references in line 50-54.
|
|
4. A short explanation governing the selection of polymers for formulation of the films should be included. |
We appreciate your excellent suggestion. The mucoadhesiveness and film-forming properties are very crucial to select polymers. We added the detailed explanation of selecting the polymers in line 89-93. |
|
5. Line 343, explain sufficient drug delivery to the buccal film. Does this mean release from the particles into the film, or is it expected that the nanoparticles diffuse intact. Is there any evidence of this? |
We are so sorry for making this confusion. We revised this sentence to make it clearer as follows in line 366-367: “To achieve optimal buccal drug delivery, it was reported that the adhesion time of the buccal film to the mucosa should be maintained within 4 to 10 h.” In addition, we added the scientific evidence in the Line 540-544 as follows: While applying LON- or d-LON loaded MBF in buccal mucosa, the nanoparticles diffused out and then experienced extensive hydrolysis by esterase in saliva or blood, liberating active LEU as reported previously by our group [20,47] . It is well known that human saliva presents esterase activity in addition to blood having abundant amount of esterase [48]. Many thanks for your excellent comments. |
|
6. There are no stability studies in the paper and the authors should comment on how any implications for storage of films and how stability could be affected. What was the water content of the different films? DSC does not seem to indicate that there is residual moisture, but how was this controlled? Given the documented stability issues with this drug (and indeed referenced in the dissolution testing), the authors could discuss any impact of modifications on the stability profile and the impact of this in formulations. |
Thank you for your insightful comments. In light of your valuable feedback, we recognize the significance of discussing potential implications for the stability of the new fattigated peptide drug formulation. The water content (LOD) of MBF formulations was invariably ranged 4.79±0.39 % after drying process. The optimal MBF formulation was stable and unchanged in a room temperature (25 oC), maintaining drug content, water content and dissolution profiles. We added these sentences in the text (Line 332-336). However, the detailed stability studies of the MBF during various storage conditions was not conducted yet by varying temperature and humidity. Therefore, future work would be carried out to give the comprehensive discussion on these aspects. We appreciate your raised concerns. In this study, the mechanistic understanding of fattigated nano peptide was mainly investigated to enhance permeability of LEU through buccal film with different strategies such as mucoadhesiveness, physicochemical properties, self-assembled nanonization of fattigated peptide, and deformable behavior of fattigated LEU nanoparticles. |

Reviewer 2 Report
Comments and Suggestions for Authors
The article is accepted without changes. It is very well written and interesting and undoubtedly contributes to the state of the art.
Congratulations to the authors.
Author Response
Response letter to reviewers’ comments:
Reviewer #2: The article is accepted without changes.
|
Comments |
Responses |
|
It is very well written and interesting and undoubtedly contributes to the state of the art. Congratulations to the authors. |
We appreciate your nice comments on our work.
|

Reviewer 3 Report
Comments and Suggestions for Authors
The current research manuscript focusing on the development of buccal films for leuprolide for enhancing permeability and bioavailability is novel and matches the journal scope. However, few sections of the manuscript need to be improved for better understanding. The authors are suggested to address the below comments:
1. Authors are suggested to modify the title of the manuscript. The current title is complex.
2. If Leuprolide is highly water soluble, what was the reason behind converting it into amorphous nature?
3. What was the molecular weight of the permeation barrier used in the study?
4. Can authors please elaborate preparation method for LOC rather than referring the previous published literature.
5. Can authors please provide more clarity for preparing MBFs? Were LOC, LON and d-LON also loaded into the MBF? Please rewrite the procedure for better understanding. Also, please correct the footnote below Table 1 (Line 148-149).
6. Section 2.4.2: Please describe the HPLC methodology.
7. Section 2.4.8: What was the nitrogen flow rate?
8. Section 2.4.9: What do authors mean by “film detached from the PB-M”? Is it separation from the film or time at which the film was dissolved? If authors mean by separation, how can it be considered as residence time? Please explain.
9. Section 2.4.10: what type of sinker was used? Do authors mean watch glass?
10. Section 2.4.11: Can authors justify why pH 6.8 PBS was used for wetting of the film and why pH 7.4 PBS was used as receiving media?
11. Section 2.4.12: Please move HPLC methodology before section 2.4.2.
12. Which formulation does authors referring to the release profiles in Figure 3b?
13. For figures, please improve the resolution or the font size.
14. Within the figure legends please also include the formulation codes. Please correct it for all the figures.
15. Section 3.3.3: Can authors please describe what the two endothermic peaks of LEU represents.
16. For dissolution figures, authors are suggested to change the y-axis legends as “%Released”
Author Response
Response letter to reviewers’ comments:
Reviewer #3: The current research manuscript focusing on the development of buccal films for leuprolide for enhancing permeability and bioavailability is novel and matches the journal scope. However, a few sections of the manuscript need to be improved for better understanding. The authors are suggested to address the below comments:
|
Comments |
Responses |
|
1. Authors are suggested to modify the title of the manuscript. The current title is complex. |
Thank you for your thoughtful suggestion. After careful consideration, we modified the title of the manuscript as follows: Nanonization and deformable behavior of fattigated peptide drug in mucoadhesive buccal films |
|
2. If Leuprolide is highly water soluble, what was the reason behind converting it into amorphous nature? |
Thank you for your excellent question. Leuprolide (LEU), in its acetate form, exhibiting a crystal habit and high water-solubility is commercially available despite being. The morphology of LEU acetate was shown in this reference (https://doi.org/10.2147/IJN.S401048). However, we challenged to conjugate oleic acid into leuprolide to obtain the amphiphilic structure that could be self-assembled into peptide nanoparticles. Therefore, the acetate form of Leu was converted into LEU to avoid any redundant conjugation with acetate as reported previously by our group in Line 126-127 (20. Park, J.; Ngo, H.V.; Jin, H.-E.; Lee, K.W.; Lee, B.-J. Hydroxyl group-targeted conjugate and its self-assembled nanoparticle of peptide drug: Effect of degree of saturation of fatty acids and modification of physicochemical properties. International Journal of Nanomedicine 2022, 2243-2260). After conjugation, it simultaneously converted LEU into amorphous state. Collectively, this approach was beneficial to optimize the druggability of LEU for improved plasma degradation and permeability in the context of our investigation.
|
|
3. What was the molecular weight of the permeation barrier used in the study? |
Thank you for your question. Unfortunately, the suppliers did not provide the specific molecular weight cut-off values for the permeation barriers used In this study, two artificial barrier membranes: PB-M and PermeaPad were used for permeability test. The PermeaPad was hand-crafted by depositing phospholipids, e.g., phosphatidylcholine, between two support sheets, e.g., cellulose hydrate, giving the PermeaPad a “sandwich” structure. Specifically, the thin, dry phospholipid layer hydrates and swells between the two regenerated cellulose layers (with an estimated cut-off size of 7–10 kDa) in Line 495-496. In contrast, Permeation-Barrier Membrane (PB-M) launched by Logan Company is a cellulose membrane (not applicable for pore size), having similar acidity to the human epidermis, which allows the transfer of novel drugs (Line 498-499). The selection of these membrane barriers was based on their widespread use and established efficacy in numerous permeation studies of new chemical entities and formulations through the gastrointestinal tract, buccal and nasal. Therefore, we believe these membranes are well-suited for our current investigation. Your kind attention would be appreciated.
|
|
4. Can authors please elaborate preparation method for LOC rather than referring the previous published literature. |
We appreciate your suggestion. Although the preparation of LOC and LON were well-established and mentioned in previous our publications (ref: 3, 20, 47), we added to elaborate preparation method in Line 127-130. Thank you for your comment.
|
|
5. Can authors please provide more clarity for preparing MBFs? Were LOC, LON and d-LON also loaded into the MBF? Please rewrite the procedure for better understanding. Also, please correct the footnote below Table 1 (Line 148-149). |
Thank you for your kind comments. To avoid any confusion, the preparation procedure was rewritten in part 2.2.2. In addition, the Table 2 (Not Table 1) was also rearranged for better understanding. The footnotes were also corrected to enhance clarity and conciseness. We appreciate your valuable comments.
|
|
6. Section 2.4.2: Please describe the HPLC methodology. |
We mentioned the detailed HPLC methodology in the manuscript in section 2.4.2. Thank you for your suggestion.
|
|
7. Section 2.4.8: What was the nitrogen flow rate? |
Thank you for your question. The flow rate was 50 mL/min. We added this information to the revised manuscript in line 256.
|
|
8. Section 2.4.9: What do authors mean by “film detached from the PB-M”? Is it separation from the film or time at which the film was dissolved? If authors mean by separation, how can it be considered as residence time? Please explain. |
We appreciate your excellent comments. As mentioned in the text, the residence time is determined from the beginning of the experiment until the point when the film is completely eroded or detached from the membrane rather than the separation. We revised this sentence in line 266.
|
|
9. Section 2.4.10: what type of sinker was used? Do authors mean watch glass? |
Thank you for your question. In this study, we used a dissolution basket sinker (40 mesh, 25.4 mm x 34.9 mm) to keep the film submerged in the media. We added the type of sinker in the revised manuscript in line 272.
|
|
10. Section 2.4.11: Can authors justify why pH 6.8 PBS was used for wetting of the film and why pH 7.4 PBS was used as receiving media? |
Thank you for your excellent question. Saliva has a pH normal range of 6.2-7.6 with 6.7 being the average pH. Although the difference of two pH was smaller, we used pH 6.8 PBS mimicking the normal saliva pH for wetting the film. In the meanwhile, pH 7.4 PBS was used as receiving media to mimic the human physiological pH. We added this information in line 286-287, 290. Thank you for your kind comments. |
|
11. Section 2.4.12: Please move HPLC methodology before section 2.4.2. |
Thank you for your comment. We moved the HPLC method in section 2.4.2 in the revised manuscript.
|
|
12. Which formulation does authors referring to the release profiles in Figure 3b? |
In figure 3b, the release profiles of these formulations were performed by varying LEU peptide concentrations in the optimal F2 formulation in Table 2. To make it clearer, we revised the sentence in line 379-381. We appreciate your valuable question.
|
|
13. For figures, please improve the resolution or the font size. |
Thank you for your comment. We revised and uploaded all figures with improved font size and higher resolution. Your kind attention would be appreciated.
|
|
14. Within the figure legends please also include the formulation codes. Please correct it for all the figures. |
According to your comment, we added the formulation code (F2) in all Figures. Thank you for your comment. |
|
15. Section 3.3.3: Can authors please describe what the two endothermic peaks of LEU represents. |
Thank you for your kind comment. The two endothermic peaks at 87.7 oC and 154.1 oC represent the glass transition temperature (Tg) and the melting point of LEU. We described this information in the revised manuscript in line 461-462.
|
|
16. For dissolution figures, authors are suggested to change the y-axis legends as “%Released” |
We revised the y-legend as “% Released” in all dissolution figures according to your guidance. Thank you for your kind comment. |
